# The reciprocal regulation between mitochondrial-associated membranes and Notch signaling in skeletal muscle atrophy

Yurika Ito[1†], Mari Yamagata[2†], Takuya Yamamoto[3,4,5], Katsuya Hirasaka[6], Takeshi Nikawa[7], Takahiko Sato[8,9,10]*

[1]Faculty of Medical Sciences, Fujita Health University, Toyoake, Japan; [2]Department of Biomedical Engineering, Faculty of Life and Medical Sciences, Doshisha University, Kyotanabe, Japan; [3]Center for iPS Cell Research and Application, Kyoto University, Kyoto, Japan; [4]Institute for the Advanced Study of Human Biology (WPI-ASHBi), Kyoto University, Kyoto, Japan; [5]Medical-risk Avoidance based on iPS Cells Team, RIKEN Center for Advanced Intelligence Project (AIP), Kyoto, Japan; [6]Organization for Marine Science and Technology, Nagasaki University Graduate School, Nagasaki, Japan; [7]Department of Nutritional Physiology, Institute of Medical Nutrition, Tokushima University Graduate School, Tokushima, Japan; [8]Department of Ophthalmology, Kyoto Prefectural University of Medicine, Kyoto, Japan; [9]Department of Anatomy, Faculty of Medicine, Fujita Health University, Toyoake, Japan; [10]International Center for Cell and Gene Therapy, Fujita Health University, Toyoake, Japan

*For correspondence: musclestemcell@gmail.com

†These authors contributed equally to this work

Competing interest: The authors declare that no competing interests exist.

**Abstract** Skeletal muscle atrophy and the inhibition of muscle regeneration are known to occur as a natural consequence of aging, yet the underlying mechanisms that lead to these processes in atrophic myofibers remain largely unclear. Our research has revealed that the maintenance of proper mitochondrial-associated endoplasmic reticulum membranes (MAM) is vital for preventing skeletal muscle atrophy in microgravity environments. We discovered that the deletion of the mitochondrial fusion protein Mitofusin2 (MFN2), which serves as a tether for MAM, in human induced pluripotent stem (iPS) cells or the reduction of MAM in differentiated myotubes caused by microgravity interfered with myogenic differentiation process and an increased susceptibility to muscle atrophy, as well as the activation of the Notch signaling pathway. The atrophic phenotype of differentiated myotubes in microgravity and the regenerative capacity of Mfn2-deficient muscle stem cells in dystrophic mice were both ameliorated by treatment with the gamma-secretase inhibitor DAPT. Our findings demonstrate how the orchestration of mitochondrial morphology in differentiated myotubes and regenerating muscle stem cells plays a crucial role in regulating Notch signaling through the interaction of MAM.

## eLife assessment

This interesting and **important** manuscript combines in vitro and in vivo experiments to investigate the reciprocal regulation between mitochondria-associated membranes and Notch signaling in skeletal muscle atrophy, with implications beyond the single subfield of muscle atrophy. The methods, data, and analyses are **solid** and broadly support the claims.

## Introduction

Skeletal muscles play a vital role in body posture, movement, and in regulating metabolism. Regular physical activity helps to maintain muscle mass, while decreased use of skeletal muscle can lead to muscle atrophy (*Bodine et al., 2001*; *Sandri et al., 2004*). This can occur in a variety of disease states, such as muscular dystrophies, as well as in conditions where movement is limited, such as long-term bed rest or injury-induced immobility, or even in microgravity conditions during spaceflight (*Gao et al., 2018*). Additionally, muscle loss is a common aspect of the aging process (*Larsson et al., 2019*; *Distefano and Goodpaster, 2018*). The mechanisms of disuse-induced muscle atrophy and the impact on regenerative capacity in skeletal muscle have been extensively studied in various animal models, including humans.

Mitochondria are essential for maintaining tissue homeostasis and supplying adenosine triphosphate (ATP) during muscle development and regeneration. This is particularly important in energy-intensive myogenic cells, such as differentiated myofibers and muscle stem cells, as they mature (*Duguez et al., 2002*; *Ryall, 2013*; *Bhattacharya and Scimè, 2020*). The shape and size of mitochondria in mature myofibers undergo drastic changes to meet the energy demands of developmental growth, the switch between slow- and fast-twitch myofibers, exercise, and aging (*Ljubicic et al., 2010*; *Wyckelsma et al., 2017*). These morphological and functional changes in mitochondria are facilitated by the processes of fission and fusion, respectively (*Youle and van der Bliek, 2012*). Mitochondrial fusion is regulated by the transmembrane GTPase proteins, Mitofusin1 (Mfn1) and Mitofusin2 (Mfn2), which are located on the outer mitochondrial membrane (*Chen et al., 2003*; *Eura et al., 2003*; *Sin et al., 2016*; *Youle and van der Bliek, 2012*).

Ablation of both Mfn1 and Mfn2 leads to embryonic lethality in mice, while muscle-specific inactivation of *Mfn1* and *Mfn2* genes results in severe mitochondrial dysfunction and muscle atrophy (*Chen et al., 2003*; *Chen et al., 2010*). Mfn2, in particular, is necessary to tether the endoplasmic reticulum (ER) to mitochondria, thereby enhancing mitochondrial energetics (*de Brito and Scorrano, 2008*; *Filadi et al., 2018*; *Ishihara et al., 2004*). However, the specific role of the mitochondria-ER complex in developmental, regenerative, and atrophic processes in skeletal muscle remains unclear. Our research has revealed that muscle atrophy is linked to Mfn2 and the Notch signaling pathway, as demonstrated by the use of MFN2-knockout human induced pluripotent stem (iPS) cells, primary human myoblasts, and mice.

Our research demonstrates that the restoration of ER-mitochondria contacts through the regulation of Notch signaling is sufficient to partially alleviate the bioenergetic defects in MFN2-deficient human iPS cells or myogenic cells. This suggests that treatment with gamma-secretase inhibitors may be a viable therapeutic option in pathological conditions in which Mfn2 is involved. These findings provide new insights into the treatment of skeletal muscle atrophy caused by mitochondrial abnormalities.

## Results

### Human primary myotubes exhibit an atrophic phenotype in a microgravity environment

To investigate the effects of microgravity on skeletal muscle atrophy, we utilized a 3D-clinorotation system to simulate microgravity conditions with human primary skeletal muscle hOM2 cells, which are derived from the orbicularis oculi (*Yamanaka et al., 2019*). We observed that the proliferation of both myogenic cells and human iPS cells decreased in microgravity using the clinorotation (*Figure 1—figure supplement 1*), and it has been reported that initial myogenic differentiation to form myotubes is suppressed by fluid motion when the clinorotation is used (*Mansour et al., 2023*). Therefore, we examined differentiated myotubes derived from primary hOM2 cells after confluence, both with and without the clinorotation (*Figure 1A*). These primary cells were mainly differentiated into *MYH7*-positive type 1 slow-twitch myotubes after 7 days (*Figure 1—figure supplement 2*), and we found that these differentiated myotubes in microgravity, when cultured for 7 days, were more fragile and tenuous in appearance than those cultured for 2 days, compared to controls (the bottom images in *Figure 1B*). We also observed an increase in the transcripts of *TRIM63* (*MuRF1*) and *FBXO32* (*Atrogen1/MAFbx*), which are markers of muscle atrophy, in these myotubes, as the duration of microgravity increased (*Figure 1C–D*). However, the expression of *MRF4* and *MYH3*, early myogenic differentiation markers, and caspase-3 and phospho-AKT, apoptotic markers, were not altered (*Figure 1C*,

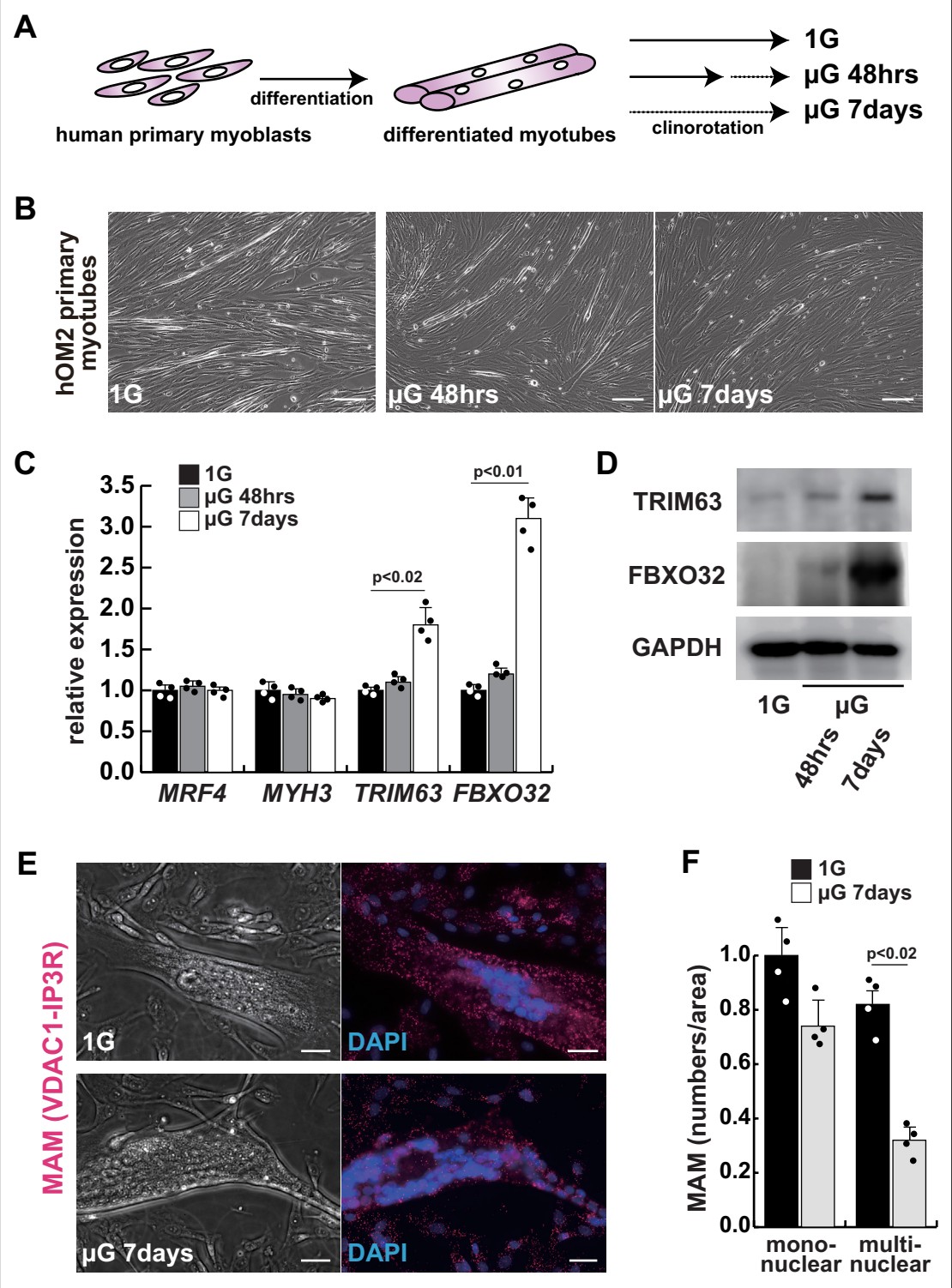

**Figure 1.** The integrity of the mitochondrial-associated endoplasmic reticulum (ER) membrane is compromised by microgravity in differentiated primary human myotubes. (**A**) A diagrammatic representation of differentiated human myotubes under normal conditions or exposed to clinorotation. μG, microgravity. (**B**) Phase contrast images of differentiated primary human myotubes under 1G (upper panel) and μG for 48 hr (left lower panel), for 7 days (right lower panel). Scale bars: 50 μm. (**C**) Transcription levels of *MRF4* (myogenic determination), *MYH3* (early differentiation), *TRIM63,* and *FBXO32* (muscle atrophy) in differentiated myotubes with or without microgravity, measured by RT-qPCR and presented relative to transcripts of ribosomal protein RPL13a. (**D**) Western blotting analyses of lysates from differentiated myotubes with or without microgravity. Nuclear lysates were analyzed with antibodies against TRIM63 (40 kDa) and FBXO32 (42 kDa). GAPDH was used as a loading control (36 kDa). (**E**) Phase contrast images and detectable

*Figure 1 continued on next page*

*Figure 1 continued*

adjacent mitochondria-associated ER membranes (MAM) by proximal ligation assay in differentiated myotube. Scale bars: 20 μm. (**F**) The number of MAM in mononuclear myoblasts (mononuclear) or multinucleated fused myotubes (multinuclear) per fixed area. All error bars indicate ± SEM (n=4). p-Values are determined by non-parametric Wilcoxon tests or one-way ANOVA and Tukey's test for comparisons.

The online version of this article includes the following source data and figure supplement(s) for figure 1:

**Source data 1.** Original western blotting images of *Figure 1D* with anti-GAPDH, anti-TRIM63, and anti-FBXO32 antibodies.

**Figure supplement 1.** The effects of microgravity on cell proliferation in human induced pluripotent stem (iPS) cells and hOM2 primary myogenic cells.

**Figure supplement 2.** Comparative expression levels of myogenic genes in differentiated human primary hOM2 myogenic cells.

**Figure supplement 3.** Apoptotic assays in differentiated hOM2 myotubes under microgravity.

**Figure supplement 3—source data 1.** Original western blotting images of *Figure 1—figure supplement 3* with anti-phospho-AKT and anti-AKT antibodies.

**Figure supplement 4.** Transcription and translational levels of Mfn1/2 expression in differentiated hOM2 myotubes under microgravity.

*Figure 1—figure supplement 3*). To investigate the condition of mitochondrial-associated endoplasmic reticulum membranes (MAM) in differentiated myotubes, we utilized a proximity ligation assay (PLA) with antibodies against the voltage-dependent anion channel 1 (VDAC1) of mitochondria and the inositol 1,4,5-triphosphate receptor (IP3 receptor) of the ER (*Prole and Taylor, 2019*; *Shoshan-Barmatz et al., 1996*). Previous studies have demonstrated that PLA can occur within a range of 40 nm, which aligns with findings suggesting that optical $Ca^{2+}$ transfer necessitates a mitochondrial proximity of approximately 20 nm (*Gottschalk et al., 2022*; *Lim et al., 2021*). We found that the number of MAM was severely decreased in microgravity, specifically in differentiated multinucleate myotubes rather than undifferentiated myoblasts (*Figure 1E–F*). It has been previously reported that the mitochondrial fusion protein MFN2 tethers MAM. We found that the protein level of MFN2 was decreased in differentiated myotubes under microgravity conditions, although the mRNA level was not significantly changed (*Figure 1—figure supplement 4*).

## Mitochondrial abnormality in human MFN2-deficient iPS cells

To investigate the role of MFN2 in atrophic myotubes differentiated from human iPS cells, we generated MFN2-mutated human iPS cells by introducing a double-strand break in the MFN2 exon3, which includes the coding sequence, using the pX458-hMFN2 editing vector with guide RNA, and two different knock-in oligos (ssODN, *Figure 2—figure supplement 1A–B*). The electroporated cells were plated on SNL feeder cells, and single cells exhibiting GFP expression were sorted through bicistronic expression with guide RNA, expanded, and subsequently, the genomic DNA was screened for the correct insertion of the knock-in reporter cassette. The genomic sequencing showed that the selected clone contained the oligo cassette (*Figure 2—figure supplement 1B*). This clone was confirmed to retain the pluripotency of undifferentiated human iPS cells using antibodies against NANOG or TRA1-81, however, the expression of MFN2 was not detected using an antibody against MFN2 (*Figure 2—figure supplement 1C–D*).

We further examined the mitochondrial condition using MFN2-deficient human iPS cells. MFN2-deficient human iPS cells showed a decrease in the number of MAM, similar to that observed in differentiated myotubes under microgravity (*Figure 2A–B*). Additionally, these cells exhibited abnormalities in mitochondrial fission (*Figure 2C*; *Cartoni and Martinou, 2009*; *Ohara et al., 2017*) and a decrease in ATP production (*Figure 2D*). To investigate the effect of MFN2 in human iPS cells, we performed next-generation sequencing (NGS) analyses comparing normal MFN2 (201B7, PB-MYOD) (*Takahashi et al., 2007*; *Tanaka et al., 2013*) to deficient MFN2 ($MFN2^{-/-}$) human iPS cells (*Figure 2—figure supplement 2*). We found that among the upregulated mRNAs expressed in MFN2-deficient iPS cells, the expression levels of Notch-related factors, such as genes of the HES or ID family, were significantly higher as shown by NGS and RT-qPCR analyses (*Figure 2—figure supplements 2–3*). In keeping with these findings, we observed that MFN2-deficient human iPS cells activated the Notch intercellular domain (NICD) in nuclei (red in *Figure 2E*). The expression levels of these upregulated factors related to Notch signaling in the MFN2-knockout condition were further increased in the microgravity condition (*Figure 2F*). To investigate the effects of MFN2 deficiency on differentiated myotubes derived from human iPS cells, these cells were induced by MYOD, activated by doxycycline (DOX) (*Sato et al., 2019*), cultured in vitro under differentiation conditions, and immunostained for MYHC expression as

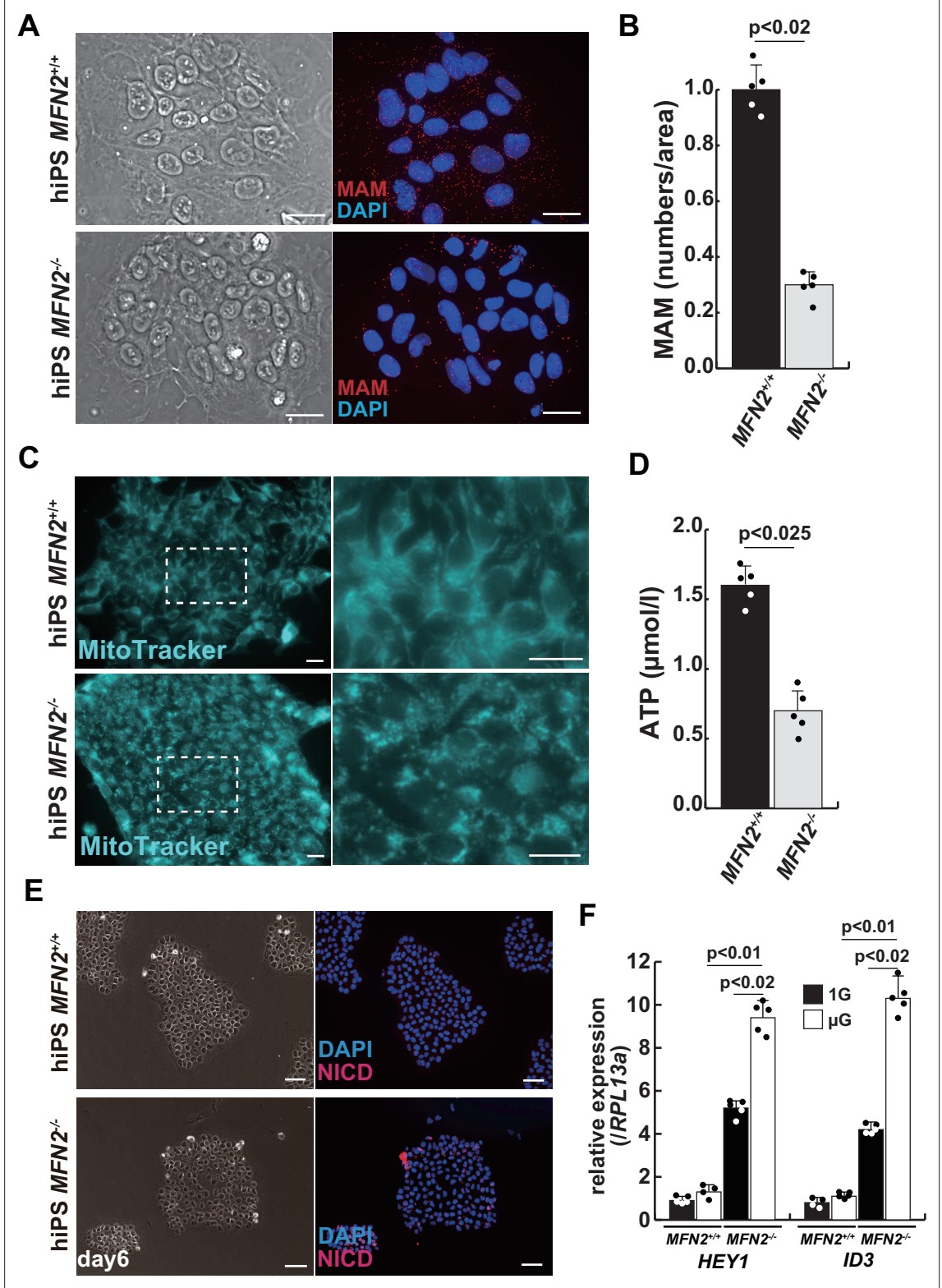

**Figure 2.** Mitochondrial abnormality and the activation of Notch in MFN2-deficient human induced pluripotent stem (iPS) cells. (**A**) Phase contrast images (left panels) and mitochondrial-associated endoplasmic reticulum membrane (MAM) visualization (right panels) with or without MFN2 in human iPS cells. Red, MAM (IP3R-VDAC1 by proximity ligation assay [PLA]); blue, DAPI. Scale bars: 20 μm. (**B**) The quantitative analyses of MAM numbers per fixed area in human iPS cells with or without MFN2. (**C**) Mitochondrial morphology with MitoTracker in human iPS cells with or without MFN2 (right panels, magnified area outlined in each left panel). Scale bars: 50 μm. (**D**) Total adenosine triphosphate (ATP) production (μmol/l) in wildtype or MFN2-

*Figure 2 continued on next page*

*Figure 2 continued*

deficient human iPS cells. (**E**) Phase contrast (left images) and immunostaining for Notch intercellular domain (NICD; red) and DAPI (blue) on wildtype or MFN2-deficient human iPS cells (right images). Scale bars: 50 μm. (**F**) Relative transcription levels of *HEY1* and *ID3* in wildtype or MFN2-deficient human iPS cells under normal gravity (1G) or microgravity (μG). All error bars indicate ± SEM (n=5). p-Values are determined by non-parametric Wilcoxon tests for comparisons.

The online version of this article includes the following figure supplement(s) for figure 2:

**Figure supplement 1.** The generation of hMFN2-deficient human induced pluripotent stem (iPS) cells.

**Figure supplement 2.** Next-generation sequencing (NGS) analyses comparing normal human induced pluripotent stem (hiPS) to deficient MFN2 (*MFN2-/-*) hiPS cells.

**Figure supplement 3.** The relative expressions of Notch-related genes in MFN2-deficient human induced pluripotent stem (iPS) cells.

**Figure supplement 4.** Differentiation of myogenic cells derived from MFN2-deficient human induced pluripotent stem (iPS) cells.

an indicator of their ability to differentiate into myotubes. We found that the number of differentiated myotubes derived from MFN2-deficient human iPS cells was significantly lower than that of wildtype cells (*Figure 2—figure supplement 4A–B*). The activation of Notch signaling was also consistent with its increase in MFN2-deficient human iPS cells (*Figure 2—figure supplement 4C*). These data indicate that the formation of differentiated myotubes derived from human iPS cells compromised in the absence of MFN2.

## The rescue of mitochondrial defects in human MFN2-deficient iPS cells by gamma-secretase inhibitor

We have confirmed that MFN2-deficient iPS cells showed abnormal mitochondrial morphology, MAM, and function. In these MFN2-deficient cells, Notch signaling was activated as recently reported in cardiomyocytes, where it was shown that the elevated Notch signaling in MFN2-deficient conditions was caused by the enzymatic activation of calcineurin (*Kasahara et al., 2013*). We observed upregulation of calcineurin activity in human iPS cells in the absence of MFN2 and in primary human muscle cells under microgravity (*Figure 3—figure supplement 1*). We therefore investigated whether the inhibition of upregulated Notch or calcineurin pathways could ameliorate mitochondrial functions in MFN2 deficiency. MFN2-deficient hiPS cells were treated with a gamma-secretase inhibitor, DAPT, and the inhibitor of calcineurin activity, FK506. We found that DAPT, but not FK506, improved mitochondrial morphology in these cells (*Figure 3A*, *Figure 3—figure supplement 2*). MFN2-deficient hiPS cells, treated with DAPT, showed decreased expression of activated HES family genes such as *HES1* and *HEY1* (*Figure 3B*). We found that the number of MAM in these cells treated with DAPT was increased (*Figure 3C–D*). Additionally, when DAPT was administrated to MFN2-deficient hiPS cells, we observed an increase in total ATP production (*Figure 3E*). These results suggest that mitochondrial functions depending on MAM are closely linked to Notch signaling.

## Gamma-secretase inhibitor DAPT ameliorates the atrophic phenotype in differentiated human myotubes under microgravity

To investigate the effect of microgravity on Notch signaling in differentiated myotubes, we evaluated Notch expression in primary human muscle cell cultures. We found that among Notch receptors, expression of the Notch2 transcript was highest in growing myoblasts and differentiated myotubes (*Figure 4—figure supplement 1*). We confirmed the activation of the NICD in both differentiated myotubes derived from MFN2-deficient human iPS cells (*Figure 4A*) and in primary differentiated myotubes, exposed to microgravity (*Figure 4B* with arrowheads). The latter also showed upregulation of Notch-related genes such as members of the HES family or ID family (*Figure 4C*) and a decline in ATP production in these cells under microgravity (*Figure 4D*).

To test the effect of inhibition of Notch signaling on the diminished mitochondrial function and increased expression of atrophic markers seen under microgravity, we administered DAPT every 2 days to cultured myotubes under microgravity (*Figure 4E*). After treatment with DAPT, we observed a significant decrease in the expression of HES family genes such as *HES1* and *HEY1* (*Figure 4F*), positive changes in mitochondrial morphology (*Figure 4G*), and partial recovery of the numbers of MAM (*Figure 4H*). Additionally, we observed an increase in total ATP production (*Figure 4I*). These results indicate that gamma-secretase inhibitors are effective in improving the mitochondrial phenotype, not

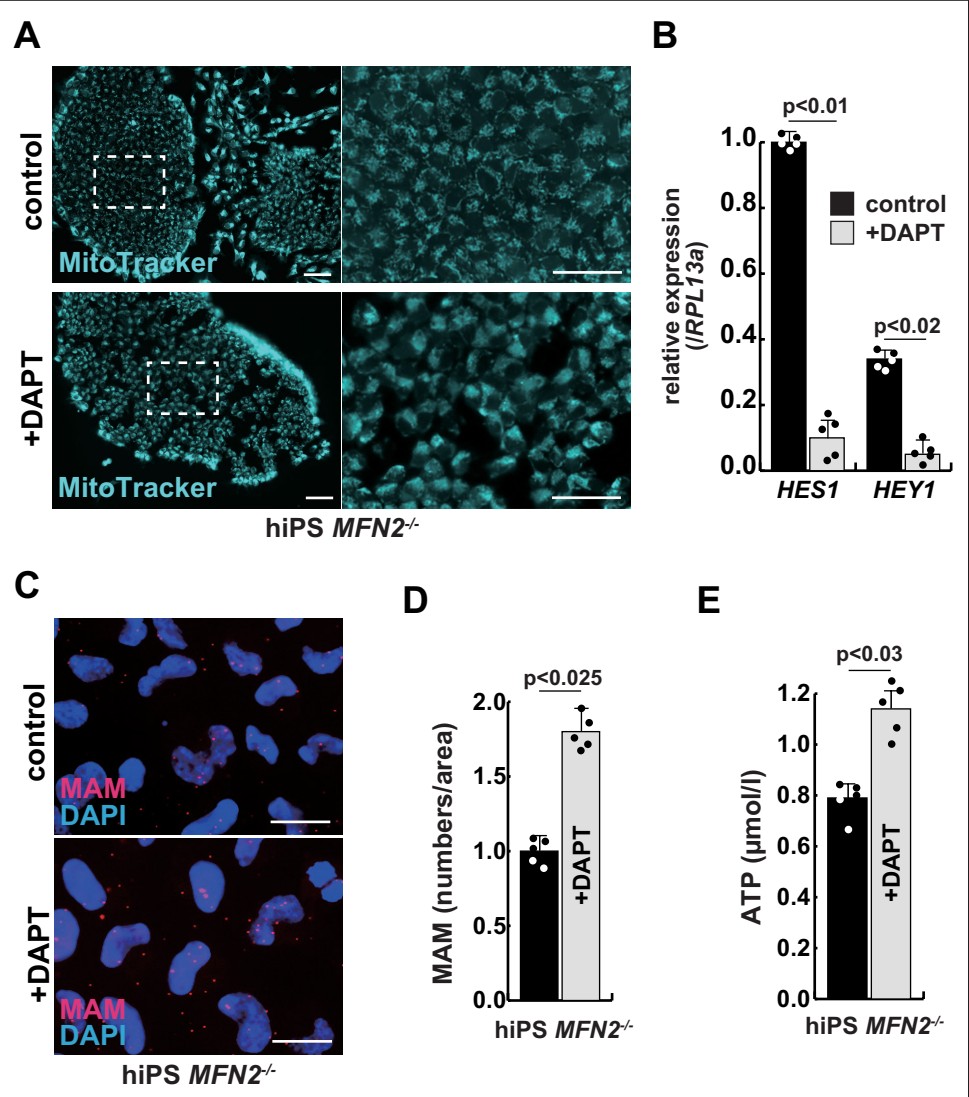

**Figure 3.** The improvement of mitochondrial abnormalities in MFN2-deficient human induced pluripotent stem (iPS) cells treated with gamma-secretase inhibitor DAPT. (**A**) Mitochondrial morphology visualized by MitoTracker in MFN2-deficient human iPS cells with or without 50 µM of DAPT (right panels, magnified area outlined in right panels). Scale bars: 50 µm. (**B**) Relative transcription levels of *HES1* and *HEY1* in MFN2-deficient human iPS cells with or without DAPT. (**C**) Mitochondrial-associated endoplasmic reticulum membrane (MAM) visualization in MFN2-deficient human iPS cells, with or without DAPT. Red, MAM (IP3R-VDAC1 proximity ligation assay [PLA]), blue, DAPI. Scale bars: 20 µm. (**D**) The quantitative analyses of MAM numbers in MFN2-deficient human iPS cells with or without DAPT. (**E**) Total adenosine triphosphate (ATP) production in MFN2-deficient human iPS cells treated with or without DAPT. All error bars indicate ± SEM (n=5). p-Values are determined by non-parametric Wilcoxon tests for comparisons.

The online version of this article includes the following figure supplement(s) for figure 3:

**Figure supplement 1.** Calcineurin activity in human induced pluripotent stem (iPS) cells and differentiated muscle cell cultures.

**Figure supplement 2.** Mitochondrial morphological changes in MFN2-deficient human induced pluripotent stem (iPS) cells remained unaltered in the presence of FK506, a calcineurin inhibitor.

only in MFN2-deficient hiPS cells but also in differentiated myotubes induced by microgravity. We also found that the treatment with DAPT led to a reduction in the level of atrophic markers such as *TRIM63* or *FBXO32* (*Figure 4J*), and a partial restoration of *MYH7* expression (*Figure 4—figure supplement 2*). These results suggest that elevated Notch signaling is casual in these adverse effects.

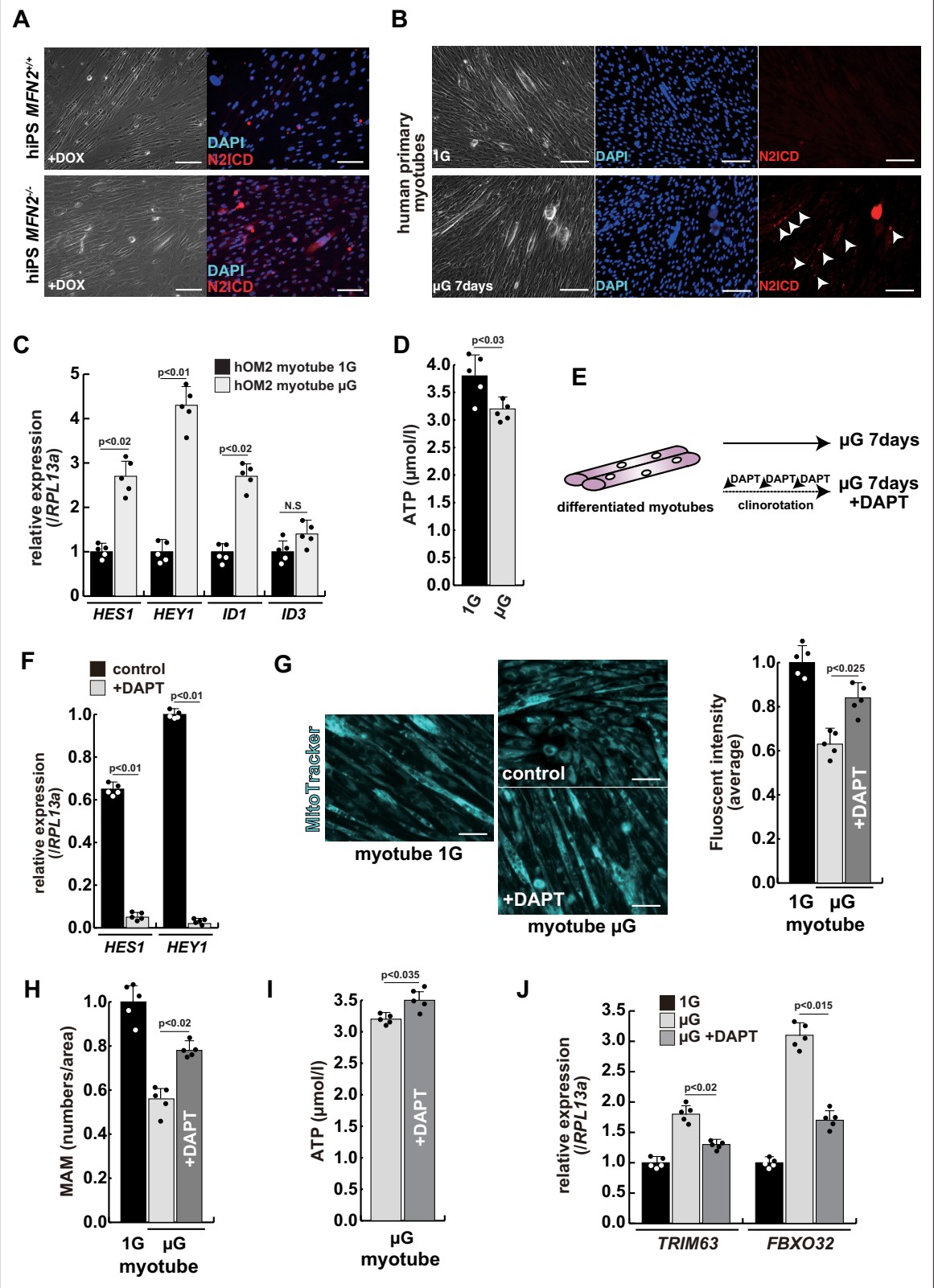

**Figure 4.** The atrophic phenotype of human myotubes under microgravity is alleviated by the gamma-secretase inhibitor DAPT. (**A**) Phase contrast (left panels) and immunostaining (right panels) for NOTCH2 intercellular domain (N2ICD; red) and DAPI (blue) on differentiated myotubes derived from wildtype, or MFN2-deficient human induced pluripotent stem (iPS) cells by doxycycline (DOX) treatment. Scale bars: 50 μm. (**B**) Phase contrast images (left panels) and immunostaining for NOTCH2 intercellular domain (right panels, N2ICD; red) and DAPI (middle panels, blue) on differentiated myotubes

*Figure 4 continued on next page*

*Figure 4 continued*

under normal gravity (1G) or microgravity (µG) for 7 days. Scale bars: 50 µm. (**C**) Relative transcription levels of HES family genes (*HES1, HEY1*) and ID family genes (*ID1, ID3*) in differentiated myotubes derived from wildtype or MFN2-deficient human iPS cells after DOX treatment. (**D**) Total adenosine triphosphate (ATP) production in differentiated human primary myotube under normal gravity (1G) or microgravity (µG) for 7 days. (**E**) The schematic representation of differentiated human primary myotubes under microgravity (µG) for 7 days with or without DAPT. (**F**) Relative transcription levels of *HES1* or *HEY1* in differentiated human primary myotubes under microgravity for 7 days with or without DAPT. (**G**) Mitochondrial morphology with MitoTracker in differentiated human primary myotubes under normal gravity (1G; left panel) and microgravity with or without DAPT (µG; upper and lower left panels). The average fluorescent intensity of total cells treated with MitoTracker is indicated on the right. Scale bars: 50 µm. (**H**) Quantitative analyses of mitochondrial-associated endoplasmic reticulum membrane (MAM) numbers in differentiated human myotubes under microgravity with or without DAPT. (**I**) Total ATP production in differentiated human myotubes under microgravity with or without DAPT. (**J**) Relative transcription levels of *TRIM63 and FBXO32* (muscle atrophy) in differentiated human primary myotubes under normal gravity (1G) and microgravity (µG) with or without DAPT for 7 days. All error bars indicate ± SEM (n=5). p-Values are determined by non-parametric Wilcoxon tests or one-way ANOVA and Tukey's test for comparisons. N.S., not significant.

The online version of this article includes the following figure supplement(s) for figure 4:

**Figure supplement 1.** Notch2 was expressed at the highest level compared to other Notch family members in human myogenic cells.

**Figure supplement 2.** Myosin heavy chain expressions in differentiated hOM2 myogenic cells with or without DAPT under microgravity.

## Muscle stem cells derived from conditional *Mfn2* mutant mice show decreased MAM and higher Notch activity, with reduced regenerative capacity of these mice after repeated injury

To further investigate the effect of Mfn2 deficiency on skeletal muscle stem cells (satellite cells) in vivo, we generated conditional Mfn2-knockout mice (*Mfn2^loxP/loxP^*; *Pax7^CreERT2/+^*), carrying an inducible Pax7-CreERT2 allele that targets muscle stem cells (*Figure 5A*). Isolated SM-C/2.6-positive muscle stem cells from these mice were treated with tamoxifen to induce Mfn2 deficiency, and then co-cultured with non-myogenic fibroblasts, as a control for muscle-specific mutation of *Mfn2* with differentiation into myotubes. We found that Mfn2 staining was present in non-myogenic fibroblasts and myotubes derived from control mice, but not in myotubes derived from Mfn2-deficient muscle stem cells (arrowheads in *Figure 5B*). As expected, the expression of Mfn2 was faint in muscle stem cells, and gradually increased in differentiated muscle fibers (*Luo et al., 2021*). Additionally, the number of MAM was decreased in differentiated myotubes derived from Mfn2-deficient muscle stem cells, as previously seen in MFN2-deficient human iPS cells or cultured myotubes under microgravity (*Figure 5C*). Furthermore, increased nuclear NICD protein was observed in cultured Mfn2-deficient muscle stem cells (*Figure 5D*). The stemness characteristics of Mfn2-deficient muscle stem cells, as indicated by Pax7 expression, were not significantly altered. However, the proportion of elongated myogenic cells in cultures of Mfn2-deficient muscle stem cells was impeded (*Figure 5—figure supplement 1*).

Next, we induced muscle injury by injecting cardiotoxin (CTX) into the tibialis anterior (TA) muscle of these conditional Mfn2-knockout mice after 4-OH tamoxifen treatment in vivo to prevent *Mfn2* expression specifically in muscle stem cells. The TA muscle of control mice was similarly injured and muscle regeneration was examined. We found that while there was no significant difference in muscle regeneration between the two groups 2 weeks after CTX injection (*Figure 5—figure supplement 2*), Mfn2-deficient mice showed a reduced capacity for muscle regeneration after repeated injury. Additionally, the regenerated muscle in Mfn2-deficient mice did not exhibit the normal hypertrophic phenotype seen in control mice (*Figure 5—figure supplement 3*; *Hardy et al., 2016*). These results suggested that Mfn2 deficiency diminishes the regenerative capacity of muscle stem cells in chronic degenerative contexts.

## The regenerative capacity of transplanted Mfn2-deficient muscle stem cells is improved by inhibition of the Notch pathway

As a model to test regeneration after cell transplantation, we used dystrophic muscle which represents a chronic degenerative context. We transplanted SM-C/2.6-positive muscle stem cells, from wildtype, or conditional Mfn2-knockout mice after 4-OH tamoxifen treatment, into the TA muscle of *Dmd^-/y^* dystrophic mice, with or without DAPT treatment (*Figure 5E*). Our findings indicated that Mfn2-deficient muscle stem cells were less capable of integrating into dystrophic muscles, as evidenced by decreased Dystrophin expression (middle panel in *Figure 5F*). However, notably, DAPT treatment

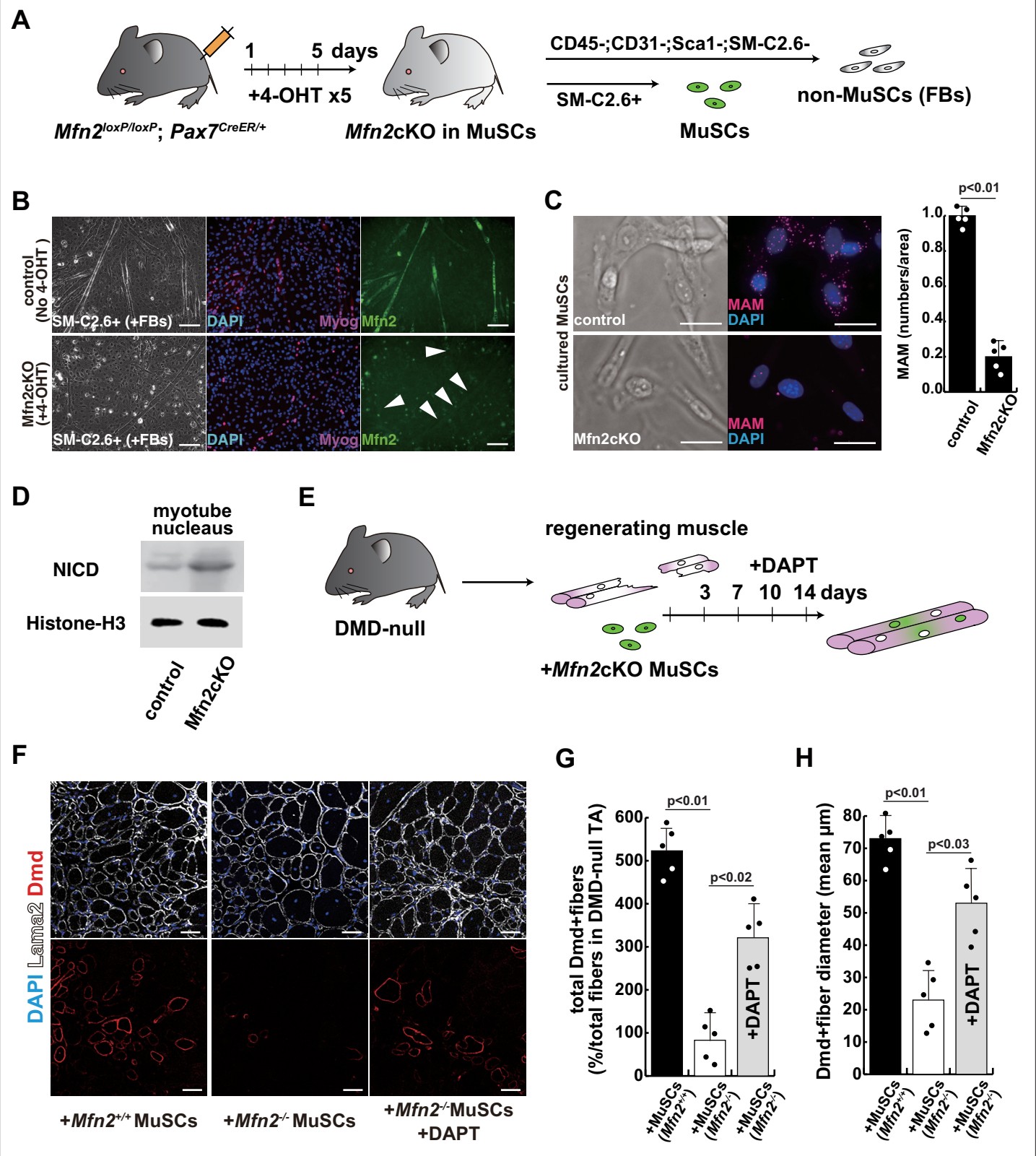

**Figure 5.** The regenerative capacity of Mfn2-deficient mouse muscle is reduced and that of Mfn2-deficient muscle stem cells (MuSCs) when transplanted into dystrophic muscle in vivo is improved by DAPT treatment of the muscle. (**A**) The flowchart to isolate MuSCs (SM-C/2.6+) and non-myogenic fibroblasts (FBs) derived from conditionally Mfn2-knockout mice after 4-OH tamoxifen (4-OHT) injection. (**B**) Immunostaining for Mfn2 (green), myogenin (Myog; red), and DAPI (blue) on differentiated myotubes derived from wildtype or Mfn2-deficient mouse MuSCs sorted as SM-C/2.6-

*Figure 5 continued on next page*

*Figure 5 continued*

positive cells, co-cultured with non-myogenic FBs. Scale bars: 50 µm. (**C**) Phase contrast images (left panels) and mitochondrial-associated endoplasmic reticulum membranes (MAMs) visualization (right panels) and quantitative analyses of MAM numbers (right) on cultured MuSCs. Red, MAM (IP3R-VDAC1 proximity ligation assay [PLA]), blue, DAPI. Scale bars: 20 µm. (**D**) Western blotting analyses of lysates from control and Mfn2-mutant cultured MuSCs. Nuclear lysates were analyzed with antibodies against Notch intercellular domain (NICD, 80 kDa). Histone H3 was used as a loading control (15 kDa). (**E**) The flowchart for the transplantation into tibialis anterior (TA) muscles of $DMD^{-/y}$ mice (12 weeks of age) with Mfn2-deficient MuSCs ($1.0\times10^4$ cells) and the treatment with DAPT every 3–4 days after the transplantation. (**F**) Transverse sectional images of TA muscles 14 days after the transplantation with the same number of MuSCs sorted as SM/C-2.6-positive cells derived from wildtype or conditional Mfn2-knockout mice. Immunostaining for Dystrophin (Dmd, red as transplanted areas), laminin-a2 (Lama2, white to show the outline of myofibers), and DAPI (blue) on engrafted TA muscle after the transplantation. Scale bars: 50 µm. (**G**) The quantification of the total number of Dystrophin-positive (Dmd+) regenerated myofibers on the section transplanted with an equivalent number of normal or Mfn2-deficient MuSCs, with or without the treatment of the transplanted muscle with DAPT. (**H**) The average diameter of Dystrophin-positive (Dmd+) myofibers that are contributed by the transplanted MuSCs, as described in (**G**). All error bars indicate ± SEM (n=5). p-Values are determined by non-parametric Wilcoxon tests or one-way ANOVA and Tukey's test for comparisons.

The online version of this article includes the following source data and figure supplement(s) for figure 5:

**Source data 1.** Original western blotting images of *Figure 5D* with anti-Histone H3 and anti-Notch intercellular domain (NICD) antibodies.

**Figure supplement 1.** Isolated muscle stem cells (MuSCs) derived from conditional *Mfn2*-knockout mice.

**Figure supplement 2.** The regenerative capacity of conditional Mfn2-knockout mice, specifically in muscle stem cells, did not exhibit any significant alteration following a single muscle injury.

**Figure supplement 3.** Reduced muscle hypertrophy in conditional *Mfn2*-knockout mice after several cardiotoxin (CTX)-induced muscle injuries.

of the TA muscles post-transplantation in this context led to increased Dystrophin expression and a larger size of regenerated myofibers (lower panel in *Figure 5F–H*).

## Discussion

Skeletal muscle atrophy is caused by various factors, including disuse or mechanical unloading. Here, we found that diminished MAM is associated with muscle atrophy in the microgravity condition. The result of our studies demonstrated the essential role of MAM in the regulation of skeletal muscle atrophy. MAM is considered to serve as a $Ca^{2+}$ transit point from the ER to mitochondria and is regulated not only by MFN2 but by the IP3R-VDAC1 protein complex (*Rizzuto et al., 1998*) which releases $Ca^{2+}$ from the ER and participates in its transfer to the mitochondria (*Szabadkai et al., 2006*). We have therefore measured the proximity of these protein complexes by the PLA method, as reported in liver or kidney cells (*Alam, 2018*; *D'Eletto et al., 2018*; *Theurey et al., 2016*).

Mfn2 is a key factor in regulating mitochondrial fusion, which is affected by the GTPase activity of both Mfn1 and Mfn2 on the mitochondrial outer membrane. It has been reported that Mfn2 but not Mfn1 independently regulates not only MAM, but also mitophagy, obesity, cardiomyopathies, and several neuronal defects such as Charcot-Marie-Tooth disease, and Parkinson's disease, suggesting that Mfn2 might not functionally compensate for Mfn1 (*Chen and Dorn, 2013*; *Filadi et al., 2018*; *Sebastián et al., 2016*). Our experiments with MFN2-disrupted iPS cells also indicated that the numbers of MAM were decreased and affected mitochondrial functions. However, no effect on developmental myogenesis or growth was observed in Mfn2-deficient mice, which showed comparable to that of wildtype mice. In muscle regeneration, Mfn2-deficient mice showed little change from wildtype mice in the event of a single muscle injury. In contrast, Mfn2 deficiency caused a muscle atrophy-like phenotype in vitro culture, however, no such muscle atrophy is observed with muscle-specific Mfn2 deficiency in vivo (*Luo et al., 2021*). One potential explanation for this discrepancy is that cell culture conditions introduce excessive mechanical stress (*Bonfanti et al., 2022*; *Valon and Levayer, 2019*). Indeed in vivo, repeated injury of muscle lacking Mfn2 did result in impaired regeneration. Future studies should consider how Mfn2 is involved in adaptation to mechanical stress.

In addition, enhanced Notch signaling was observed accompanying the decrease of MAM in MFN2-deficient iPS cells and myocytes. Reports on cardiac muscle cells suggest that there is cross-talk between $Ca^{2+}$/calcineurin with MAM and Notch signaling pathways, which may function in addition to, or in parallel with, Notch1 regulation by extracellular calcium and sarcoplasmic reticulum $Ca^{2+}$ adenosine triphosphatase (*Rand et al., 2000*; *Roti et al., 2013*; *Song et al., 2022*). Indeed, we report that when human iPS cells in the absence of MFN2 were treated with inhibitors of calcineurin and Notch signaling, the mitochondrial distribution was not significantly altered by FK506, a

calcineurin inhibitor, but by DAPT, a Notch signaling inhibitor (*Figure 3—figure supplement 2*). The same results were obtained in skeletal muscle cells, where changes in mitochondrial distribution were observed and partial functional recovery was demonstrated upon suppression of Notch signaling. Further research will be necessary to unravel the underlying mechanisms that lead to enhanced Notch signaling in MFN2 deficiency, to explore upstream pathways, other than calcium, and to elucidate the reverse control mechanism by which Notch regulates MAMs.

The Notch signaling pathway is an evolutionarily conserved cascade that plays a role in organ development and morphogenesis, stem cell fate, tissue metabolism, and in various diseases (*Andersson et al., 2011*). In skeletal muscles, Notch signaling is regulated throughout multiple stages of development and regeneration. It is essential for maintaining the dormant state and also for self-renewal of muscle stem cells in the adult (*Gioftsidi et al., 2022*; *Liu et al., 2018*). The positioning of these stem cells on muscle fibers depends on basal lamina assembly controlled by Notch (*Bröhl et al., 2012*), which also acts as an inhibitor of premature differentiation by repressing the expression of MyoD, a myogenic regulatory factor (*Delfini et al., 2000*). However, its role in myogenesis at late stages or in mature myotubes remains unclear. In studies involving the forced expression of NICD in differentiating muscle cells, Notch signaling has been shown to inhibit their maturation after myocyte fusion, but also to ameliorate the pathophysiology of mature muscle fibers in aged or dystrophic muscle (*Bi et al., 2016*). The differentiated human muscle cell culture used in this experiment reported here do not form fully mature fibers. Therefore, the effects of Notch inhibition by DAPT at later stages, using more highly organized culture systems to prompt maturation, require further investigation.

Activated Notch signaling has been reported in dystrophic or atrophic muscles (*Fujimaki et al., 2022*; *Sakai-Takemura et al., 2020*) and also in other pathological situations such as neuroblastomas (*Ferrari-Toninelli et al., 2010*) or in ischemic stroke where inhibition of Notch signaling with DAPT is beneficial (*Balaganapathy et al., 2018*). Our study has shown that treatment of DAPT can counteract the negative effect of MFN2 deficiency in human iPS cells and myocytes. Furthermore, grafting efficiency of Mfn2-deficient muscle stem cells is improved if the recipient muscle is treated with DAPT. These experiments point to the positive effects of inhibiting Notch in this context. Importantly, reduction in Notch signaling by DAPT addition in muscle cells under microgravity resulted in higher Mfn2 and MAM levels with improved mitochondrial function and reduced atrophy. This opens the possibility that compounds to inhibit Notch signaling may be of therapeutic value in treating pathological muscle wasting. The treatment of various cancers and of Alzheimer's disease, in which the Notch pathway has been implicated, is in progress (*McCaw et al., 2021*; *You et al., 2023*). Further investigation will be required to find out whether these gamma-secretase inhibitors are also effective against muscle atrophy and muscle diseases.

## Methods
### Primary human myogenic cell culture
All methods relating to human studies were performed in accordance with the guidelines and regulations of Medical Research Ethics Review Committee of Fujita Health University (approved number CI22-087). Informed consent was obtained prior to ophthalmic blepharoplasty surgery, allowing for publication elsewhere. Human biopsies of the extra eyelid tissue from a young male, including orbicularis oculi muscle, were minced and subjected to enzymatic dissociation with 0.1% Collagenase Type2 (Worthington) in DMEM (WAKO) at 37°C for 60 min. Dissociated cells from biopsies or sorted cells were plated in DMEM containing 20% FBS and 5 ng/ml of basic FGF (WAKO, Osaka, Japan), coated with Geltrex (GIBCO). Fresh media was added regularly until colonies with spindle-shaped cells were obtained. For primary myogenic cell sorting, expanded cells were detached with Accutase (Nacalai tesque) from cell culture dishes, resuspended with 1% bovine serum albumin (Sigma-Aldrich) in PBS buffer (WAKO), and incubated with the monoclonal anti-human antibodies anti-CD56-PE and anti-CD82-Alexa647 (BioLegend). After 30 min incubation on ice, human myoblasts including muscle stem cells, defined as CD56+CD82+, were sorted by flow cytometry using a FACS JAZZ (BD). Isotype control antibodies were PE- and Alexa647-conjugated mouse IgG1 (BioLegend), filtrated with a cell strainer (35 μm, BD). Cell suspensions were stained with SYTOX Green Dead Cell Stain (Molecular Probes) to exclude dead cells. These primary human myogenic cells are referred to as hOM2 cells. These hOM2 myogenic cells were cultured in DMEM containing 20% of FBS and 5 ng/ml of basic FGF.

After a few days of cell culture at 70–80% of confluency, these cells were differentiated into myotubes in DMEM supplemented with 2% horse serum (GIBCO). All cells were verified to be free of mycoplasma by MycoAlert PLUS (Lonza).

## Human iPS cell culture

KYOU-DXR0109B human iPS cells (ATCC; 201B7) (*Takahashi et al., 2007*) were cultured on 0.1% of Gelatin-coated dishes in Primate ES cell medium (Reprocell) supplemented with 5 ng/ml of basic FGF with SNL76/7 feeder cells (ATCC; SCRC-1049), or iMatrix (Nippi)-coated dishes in StemFit AK02 medium (Ajinomoto) without feeder cells. Human iPS cells were passaged as the condition of single cells. The derivation of myogenic cells from hiPS cells based on MYOD1 induction (*Sato et al., 2019*; *Tanaka et al., 2013*) was followed. Single iPS cells carrying an inducible MYOD1 activation system were expanded in Primate ES cell medium (Reprocell) without bFGF and with 10 µM of Y-27632 (Nacalai tesque) for 24 hr, and then induced into myogenic cells by adding 500 ng/ml of DOX (Tocris). After 24 hr, the cell culture medium was changed into myogenic differentiation medium composed of alpha-MEM (Nacalai tesque) with 5% of KSR (GIBCO) and 500 ng/ml of Dox. After 6 days, the culture medium was changed into muscle maturation medium, DMEM/F12 with 5% of horse serum, 10 ng/ml of recombinant human insulin-like growth factor 1 (Peprotech), and 200 µM of 2-mercaptoethanol (Sigma-Aldrich). All cells were authenticated by STR profiling (ATCC) and were verified to be free of mycoplasma.

## hMFN2 targeting with human iPS cells

To introduce into cultured cells with hMFN2-targeting vector by CRISPR/Cas9 system, which was constructed with pX458 vector (Addgene #42230) by ligating oligos into it (MFN2 exon3 target site: CAGTGACAAAGTGCTTAAGT), and synthesized oligos for knock-in (50mer arm+1bp+50 mer arm: CAGTCAAGAAAAATAAGAGACACATGGCTGAGGTGAATGCATCCCCACT-t/c-TAAGCACTTTGT CACTGCCAAGAAGAAGATCAATGGCATTTTTGAGCAGC), the electroporator NEPA21 (NEPAGENE) was used for introducing plasmid DNAs into hiPS cells as described (*Sato et al., 2019*). Cultured hiPS cells transfected with pX458-*hMFN2* were dissociated with TrypLE select (GIBCO) at 37°C for 5 min for detecting transfected cells. Dissociated cells were resuspended with 1% bovine serum albumin in PBS. Cell debris was eliminated with a cell strainer (35 µm), and dissociated cell suspensions were centrifuged and stained with SYTOX Red Dead Cell Stain (Molecular Probes) to exclude dead cells. Single GFP-positive cells on the 96-well cell culture plate were collected by FACSJAZZ. Sorted single cells were expanded for a few weeks to analyze genomic DNA.

## 3D-Clinostat

3D-Clinostat (Zeromo CL-5000, Kitagawa Corporation) was used to produce microgravity conditions by rotating a sample around two axes, creating a pseudo-environment similar to that of outer space (1/1000G). Differentiated human myotubes or human iPS cells were cultured in cell culture dishes (35 mm, Corning) for 48 hr or 7 days in simulated microgravity, while cells in the same dishes were cultured under 1G ground conditions as a control. The use of a gravity acceleration sensor helped to define the simulated microgravity conditions within a few minutes. A detailed description of the metabolic analyses with myogenic cells cultured in microgravity conditions will be published elsewhere (Sugiura et al., in preparation).

## RNA-seq analysis

Total RNAs from cultured human iPS cells were extracted using NucleoSpin RNA Plus XS (Macherey-Nagel). The 100 ng of total RNAs were used as starting materials to generate RNA-seq libraries with the TruSeq Stranded mRNA LT sample prep kit (Illumina). The obtained libraries were sequenced on a NextSeq500 (Illumina) as 75 bp single-end reads. After trimming adaptor sequences and low-quality bases with cutadapt-1.18, the sequenced reads were mapped to the mouse reference genome (mm10) with STAR v 2.6.0c, with the GENCODE (release 36, GRCh38.p13) gtf file. The raw counts for each gene were calculated using htseq-count v0.11.2 with the GENCODE gtf file. Gene expression levels were determined as transcripts per million and differentially expressed genes were identified with DESeq2 v1.30.1. Raw data concerning this study were submitted under Gene Expression Omnibus (GEO) accession number GSE226330.

## RT-qPCR analyses

Total RNAs from sorted or cultured cells were extracted using NucleoSpin RNA Plus XS. For quantitative PCR analyses, synthesized cDNAs were prepared using SuperScript VILO MasterMix (Invitrogen). All RT-qPCRs were carried out in triplicate using ThunderBird SYBR qPCR Mix (TOYOBO) and Thermal Cycler Dice Realtime System (TAKARA), and normalized to mRNA expression level of human *ribosomal protein L13A, GAPDH, and ACTB* as controls. Primer sequences (5′ to 3′) are listed in *Supplementary file 1*.

## Western blot

The cells were lysed with radio-immunoprecipitation assay buffer (Nacalai tesque) or ProteoExtract Subcellular Proteome Extraction Kit (Millipore) containing a protease inhibitor cocktail (Nacalai tesque). The supernatant containing the total proteins was fractionated after centrifugation by sodium dodecyl sulfate-polyacrylamide gel electrophoresis (TEFCO). The separated proteins were transferred to polyvinylidene difluoride membranes (TEFCO), blocked with 5% of BlockingOne (Nacalai tesque) for 30 min, and incubated with anti-MFN2 (diluted 1/1000, Cell Signaling Technology), anti-TRIM63 (diluted 1/2000, GeneTex), anti-FBXO32 (diluted 1/2000, Proteintech), anti-NICD (diluted 1/1000, Cell Signaling Technology), anti-AKT (pan, diluted 1/1000, Cell Signaling Technology), anti-Phospho-AKT (Ser473, diluted 1/2000, Cell Signaling Technology), anti-Histon H3 (diluted 1/1000, Abcam), anti-Hsp601 (HSPD1, diluted 1/1000, Abcam), and anti-GAPDH (diluted 1/1000, Abcam) primary antibodies overnight at 4°C. The blots were probed with horseradish peroxidase-conjugated secondary antibodies (Molecular Probes; diluted 1/5000) and developed with luminal for enhanced chemiluminescence using Chemi-Lumi One Super (Nacalai tesque). When probing for multiple targets, a single membrane was stripped with WB Stripping Solution (Nacalai tesque) and re-probed with antibodies again.

## Immunofluorescence

Cultured cells were fixed with 4% paraformaldehyde in PBS for 15 min at 4°C. Subsequently, the samples were incubated with 0.1% TritonX-100 in PBS for 5 min and then blocked with BlockingOne for 30 min. The cells were then incubated overnight at 4°C with a variety of primary antibodies including anti-MFN2 (Cell Signaling Technology; diluted 1:200), anti-TRA-1–81 (diluted 1:200, Cell Signaling Technology), anti-NANOG (diluted 1:200, Cell Signaling Technology), anti-NICD (diluted 1/200, Abcam), anti-NOTCH2 ICD (N2ICD, dilute 1/200, R&D), anti-PAX7 (diluted 1/100, DSHB), anti-Myogenin (diluted 1/100, DAKO), anti-MYH3 (diluted 1/100, DSHB), anti-Laminin 2a (diluted 1/500, Enzo Life Sciences), anti-Dystrophin (diluted 1/200, Abcam), anti-Caspase3 (diluted 1/100, Proteintech) antibodies in 5% of BlockingOne in PBS with 0.1% Tween20 (PBST). Following three washes with PBST, the cells were incubated with Alexa-conjugated anti-mouse, rabbit, or rat IgG antibodies (Molecular Probes; diluted 1/500). The cells were then washed and mounted in ProLong Diamond antifade reagent with DAPI (Molecular Probes) and images were collected and processed using a BZX-700 microscopy (Keyence).

## Visualized MAMs with PLA

Fixed cells with 4% PFA solution washed with PBS and blocked with BlockingOne for 30 min at room temperature. Next, anti-IP3R (diluted 1/200, Abcam) and anti-VDAC1 (diluted 1/200, Abcam) primary antibodies were incubated overnight at 4°C. Following washing with PBS, MAM was visualized using Duolink PLA Reagents (Sigma-Aldrich). Secondary antibodies conjugated with anti-mouse PLUS and anti-rabbit MINUS oligonucleotides were incubated for 1 hr at 37°C. After a 30 min reaction at 37°C using Ligation solution mixed with 5×Ligation stock and Ligase, the sample was incubated with Amplification solution, containing 5×Amp Red solution and polymerase, for 100 min at 37°C. MAM signals, visualized as red particles, were quantified using a hybrid cell counting system (Keyence).

## Enzymatic activity assay

The intracellular ATP levels were determined using the luciferase method with ATP Assay Kit-Luminescence (DOJINDO) as per the manufacturer's instructions. Briefly, collected cells were washed with PBS, and lysed using an attached Assay Buffer. The supernatant was incubated with a freshly prepared Working solution containing substrate and enzyme solution and then subjected to

bioluminescent detection using a microplate reader (Infinite 200 PRO, TECAN). The ATP level was measured as a control with 1 μmol/l of ATP stock solution. Calcineurin activity was measured using a cellular calcineurin phosphatase activity assay kit (Abcam), following the manufacturer's instruction by the microplate reader.

### Mouse lines

All animal experiments were approved by the ethics committee of Animal Experimentation of Fujita Health University (permission number AP18019). Needle injections were performed under anesthesia, and all efforts were made to minimize suffering. $MFN2^{loxP/+}$ (Jackson Laboratory #026525), $Pax7^{CreERT2/+}$ (Jackson Laboratory #017763) mice were used to obtain skeletal muscle cells, 4-hydroxytamoxifen (4-OHT; Sigma-Aldrich) was administrated to 12-week-old male mice at a dose of 1.0 mg/40 g body weight via intraperitoneal injection for 5 consecutive days. Female $Dmd^{-/y}$ mice crossed with male NSG mice (Charles River) (*Sato et al., 2014*) were used as transplanted donors. Male $Dmd^{-/y}$; NSG mice were used for all experiments at the indicated ages.

### Mouse muscle stem cells sorting

For the isolation and culture of live skeletal muscle stem cells in mice, TA muscles were treated with 0.1% Collagenase Type2 in DMEM/F12 at 37°C for 60 min. The dissociated cells were resuspended with 1% BSA in PBS and filtrated through a cell strainer. The cell suspensions were stained with anti-SM-C/2.6 antibody (*Fukada et al., 2004*) as well as anti-CD45-PE (diluted 1/500, BioLegend), anti-CD31-PE (diluted 1/500, BioLegend), anti-Sca1-PE (diluted 1/500, BioLegend) antibodies to exclude non-muscle cells.

### Grafting of muscle stem cells into TA muscles of DMD mice

$Dmd^{-/y}$; NSG host male mice aged 12 weeks were used for engraftment of freshly isolated muscle stem cells derived from wildtype or conditional Mfn2-knockout mice ($1.0 \times 10^4$ cells per 20 μl of PBS) into TA muscles. TA muscle was removed 2 weeks after transplantation with several injections of DAPT solution (20 μl of 50 μM stock, WAKO) as shown in *Figure 5E*, fixed, and stained as above. For quantification, serial transverse sections were cut across the entire TA muscle, generating approximately 20 slides per muscle, each containing about 20 serial sections. Five distinct slides were immunostained, encompassing regions where the majority of engrafted cells were located, to quantify the number of Dystrophin-positive (donor cells) or Lama2-positive (whole) myofibers using a hybrid cell counting system. At least four transplanted mice were analyzed per experiment.

### Statistics

We present statistical data, including the results of multiple biological replicates. The statistical analyses were conducted using Prism9 software (GraphPad Software), employing non-parametric Wilcoxon tests to compare two factors, and one-way ANOVA followed by Tukey's comparison test to determine significant differences among more than three factors. The p-values are indicated on each figure and considered significant when <0.05. All error bars are represented as means ± SEM unless otherwise stated.

## Acknowledgements

This work was supported by the Japan Society for the Promotion of Science, KAKENHI grants 17K01859, 18H04061, 23K10971, Japan Agency for Medical Research and Development (AMED) CREST 21gm0910009h0506, JP16gm0810009, the Nakatomi Memorial Foundation, and the Hori Sciences and Arts Foundation. The author declared no conflict of interest.

## Additional information

### Funding

| Funder | Grant reference number | Author |
|---|---|---|
| Japan Society for the Promotion of Science | 17K01859 | Takahiko Sato |
| Japan Society for the Promotion of Science | 18H04061 | Takahiko Sato |
| Japan Society for the Promotion of Science | 23K10971 | Takahiko Sato |
| Japan Agency for Medical Research and Development | 21gm0910009h0506 | Takahiko Sato |
| Japan Agency for Medical Research and Development | JP16gm0810009 | Takahiko Sato |
| Nakatomi Memorial Foundation | | Takahiko Sato |
| Hori Sciences and Arts Foundation | | Takahiko Sato |

The funders had no role in study design, data collection and interpretation, or the decision to submit the work for publication.

### Author contributions

Yurika Ito, Mari Yamagata, Data curation, Formal analysis, Validation, Investigation, Visualization; Takuya Yamamoto, Data curation, Software, Formal analysis, Validation; Katsuya Hirasaka, Data curation, Formal analysis, Visualization; Takeshi Nikawa, Conceptualization, Resources, Supervision; Takahiko Sato, Conceptualization, Formal analysis, Supervision, Funding acquisition, Validation, Investigation, Visualization, Writing – original draft, Project administration, Writing – review and editing

### Author ORCIDs

Katsuya Hirasaka ⬚ http://orcid.org/0000-0003-2645-8450
Takahiko Sato ⬚ https://orcid.org/0000-0003-3836-7978

### Ethics

All methods relating to human studies were performed in accordance with the guidelines and regulations of Medical Research Ethics Review Committee of Fujita Health University (approved number CI22-087). Informed consent was obtained prior to ophthalmic blepharoplasty surgery, allowing for publication elsewhere.
All animal experiments were approved by the ethics committee of Animal Experimentation of Fujita Health University (Permission number AP18019).

Reviewer #1 (Public Review): https://doi.org/10.7554/eLife.89381.3.sa1
Reviewer #2 (Public Review): https://doi.org/10.7554/eLife.89381.3.sa2
Author Response https://doi.org/10.7554/eLife.89381.3.sa3

## Additional files

### Supplementary files

• Supplementary file 1. Primers for the expression analysis by RT-qPCR of the mRNAs are indicated.
• MDAR checklist

### Data availability

Sequencing data have been deposited in GEO under accession number GSE226330.

The following dataset was generated:

| Author(s) | Year | Dataset title | Dataset URL | Database and Identifier |
|---|---|---|---|---|
| Sato T | 2023 | RNA-seq analysis with MFN2-deficient human iPS cells | http://www.ncbi.nlm.nih.gov/geo/query/acc.cgi?acc=GSE226330 | NCBI Gene Expression Omnibus, GSE226330 |

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
