## [Editor Report · eLife assessment]

This interesting and **important** manuscript combines in vitro and in vivo experiments to investigate the reciprocal regulation between mitochondria-associated membranes and Notch signaling in skeletal muscle atrophy, with implications beyond the single subfield of muscle atrophy. The methods, data, and analyses are **solid** and broadly support the claims.

---

## [Referee Report · Reviewer #1 (Public Review)]

In this study, the authors investigated the role of MAM and the Notch signalling pathway in the onset of the atrophic phenotype in both in vivo and in vitro models. The rationale used to obtain the data is one of the main strengths of the study. Already from the reading, the reasoning scheme used by the authors in setting up the study and evaluating the data obtained is clear. Using both cellular and mouse models in vivo consolidates the data obtained. The authors also methodologically described all the choices made in the supplementary section.

---

## [Referee Report · Reviewer #2 (Public Review)]

In this study, the authors examined how maintenance of mitochondrial-associated endoplasmic reticulum membranes (MAM) are critical for the prevention of muscle atrophy under microgravity conditions. They observed, a reduction in MAM in myotubes placed in a microgravity condition; in addition, MFN2-deficient human iPS cells showed a decrease in the number of MAM, similar to in myotubes differentiated under microgravity conditions, in addition to the activation of the Notch signaling pathway. The authors, morover, obsreved that by treatment with the gamma-secretase inhibitor with DAPT preserved from the atrophic phenotype of differentiated myotubes in microgravity and improve the regenerative capacity of Mfn2-deficient muscle stem cells in dystrophic mice.

The entire study was well conducted, bringing an interesting analysis in vitro and in vivo of aging condition. In my opinion it is necessary to implement the analysis of both genes and proteins for better supporting the conclusions

The study can contribute to better understand one of the major problems of aging, such as muscle atrophy and inhibition of muscle regeneration, emphasizing the importance of NOTCH patway in these pathological situations. The work will be of interest to all scientist working on aging.

---

## [Author Response]

The following is the authors’ response to the original reviews.

**Reviewer #1 (Public Review):**
In this study, the authors investigated the role of MAM and the Notch signaling pathway in the onset of the atrophic phenotype in both in vivo and in vitro models. The rationale used to obtain the data is one of the main strengths of the study. Already from the reading, the reasoning scheme used by the authors in setting up the study and evaluating the data obtained is clear. Using both cellular and mouse models in vivo consolidates the data obtained. The authors also methodologically described all the choices made in the supplementary section. A weakness, on the other hand, is the failure to include averages and statistical data in the results that would give a quantifiable idea of the data obtained. To complete the picture, the authors could also investigate the possible involvement of the intrinsic apoptosis pathway as well as describe probable metabolic shifts to muscle cells in atrophic conditions. The rationale used by the authors to obtain the result is linear. The data obtained are useful for understanding the onset and characterization of the atrophic phenotype under disuse and microgravity conditions. The methods used are in line with those used in the field and can be a starting point for other studies. The cellular models are well described in the Materials and methods section. The selected mouse models followed a logical rationale and were in line with the intended aim.

We thank this reviewer for comments that have led us to clarify several points.

**Reviewer #1 (Recommendations For The Authors):**
In order to reinforce and justify the results obtained, I would suggest that the authors include numerical and statistical data in the results obtained.

Answer As the reviewer suggested, we have incorporated actual numerical and statistical data into each graph in all figures.

With the aim of better framing the picture of an atrophic muscle phenotype caused by microgravity or disuse, I would advise the authors to also focus on the possible involvement of the intrinsic apoptosis pathway. To this end, it would be interesting to assess a possible relationship between MAM and apoptosis. It would be useful to integrate this part into the discussion.

Answer It has been shown that suppression of Mfn2 expression attenuates calcium influx into mitochondria during apoptosis-inducing stimuli, thereby inhibiting apoptosis (Martins de Brito & Scorrano, Nature 2008), however, in our study, we found that apoptotic pathways, including Caspase3 or p-AKT were not significantly altered in differentiated human myocytes by microgravity for 7 days in culture, suggesting that microgravity-induced apoptosis is not an initial pathway to MAM. We have added these data in the new supplementary file 3 and mentioned it in the results.

In addition to TA, did the authors investigate what was seen in other muscles impacted by microgravity? If so, I would recommend supplementing what is available or, on the contrary, justifying the exclusivity of the choice of TA.

Answer It has been reported that the soleus, a slow-type muscle is more susceptible than the fast-type tibialis anterior muscle during gravity changes, so it makes more sense for the content of this study to analyze the soleus muscle. However, we chose the tibialis anterior muscle as our target because it provides the most stable results as a site for stem cell transplantation to observe muscle regeneration.

The authors affirm that there is an altered distribution and morphology of mitochondria under microgravity conditions. To corroborate this assertion, I would recommend including a morphological image that confirms it.

Answer The morphology of mitochondria in cultured myotubes, as observed by mitotracker staining in Figure 4G, varied widely, from finely divided to fused even within a single fiber compared to MFN2-mutated human iPS cells, making it difficult to conclude whether these changes were brought about by microgravity. Therefore, in this study, we have shown that they are reduced in microgravity by the difference in fluorescence intensity of mitotracker, which is directly proportional to mitochondrial activity.

It would be interesting if the authors would show whether there are changes in myosin expression or metabolic changes in cells subjected to microgravity and in the cell model with Mnf2 deletion. It would also be interesting to evaluate this in the presence of DAPT.

Answer As the reviewer’s suggestion, we have checked MYH1, MYH3, and MYH7 transcripts in differentiated myotubes under microgravity, with or without DAPT in the new supplementary file 12. We have added the data showing that not MYH1 but MYH7 transcript was partially recovered in the Results.

A detailed description of the metabolic analyses with myogenic cells cultured in microgravity conditions will be published elsewhere (Sugiura et al., “Mitochondria aconitase is a main target for unloading-mediated mitochondria dysfunction toward muscle atrophy”, in preparation). We have described it in the Materials and methods of the manuscript.

**Reviewer #2 (Public Review):**
In this study, the authors examined how the maintenance of mitochondrial-associated endoplasmic reticulum membranes (MAM) is critical for the prevention of muscle atrophy under microgravity conditions. They observed, a reduction in MAM in myotubes placed in a microgravity condition; in addition, MFN2-deficient human iPS cells showed a decrease in the number of MAM, similar to in myotubes differentiated under microgravity conditions, in addition to the activation of the Notch signaling pathway. The authors, moreover, observed that treatment with the gamma-secretase inhibitor with DAPT preserved the atrophic phenotype of differentiated myotubes in microgravity and improve the regenerative capacity of Mfn2-deficient muscle stem cells in dystrophic mice. The entire study was well conducted, bringing an interesting analysis in vitro and in vivo of aging conditions. In my opinion, it is necessary to improve the analysis of both genes and proteins to better support the conclusionsThe study can contribute to a better understanding of one of the major problems of aging, such as muscle atrophy and inhibition of muscle regeneration, emphasizing the importance of the NOTCH pathway in these pathological situations. The work will be of interest to all scientists working on aging

We thank this reviewer for the positive comments and remarks that we have attempted to address.

**Reviewer #2 (Recommendations For The Authors):**
Results:In Figure 1b authors observed an increase in the transcripts of MuRF1 and FBXO32 after 7 days of microgravity condition. I suggest to investigate the protein expression of these genes to give more validation to this data.

Answer As the reviewer’s suggestion, we have investigated the western blotting with atrophic markers in microgravity samples. These data have been added in Figure 1D.

Moreover, I suggest investigating not only Myogenin as an earlier gene of myotubes formation but also MRF4.Methods:I suggest when doing real-time PCR not to use a single gene as housekeeping but the average of three genes, to avoid the influence of a single housekeeping gene affecting the results.

Answer As the reviewer’s suggestion, we have investigated MRF4 expression by qPCR experiments with 3 different housekeeping genes (RPL13a, GAPDH, and ACTB). Our experiments showed no significant differences among these three housekeeping genes.We have added these data to Figure 1C and Methods in the manuscript.